# Effect of Physical and Enzymatic Pre-Treatment on the Nutritional and Functional Properties of Fermented Beverages Enriched with Cricket Proteins

**DOI:** 10.3390/foods10102259

**Published:** 2021-09-23

**Authors:** Chaima Dridi, Mathieu Millette, Blanca Aguilar, Johanne Manus, Stephane Salmieri, Monique Lacroix

**Affiliations:** 1INRS Armand-Frappier Health Biotechnology Research Centre, Research Laboratories in Sciences, Applied to Food (RESALA), Canadian Irradiation Centre (CIC), Institute of Nutrition and Functional Foods (INAF), 531 Boulevard des Prairies, Laval, QC H7V 1B7, Canada; Chaima.Dridi@inrs.ca (C.D.); Johanne.Manus@inrs.ca (J.M.); Stephane.Salmieri@inrs.ca (S.S.); 2Bio-K Plus International Inc., a Kerry Company, Preclinical Research Division, 495 Armand-Frappier Blvd, Laval, QC H7V 4B3, Canada; mmillette@biokplus.com; 3Research Laboratory of Industrial Microbiology, Centro Universitario de Ciencias Exactas e Ingenierías, Universidad de Guadalajara, 1421, Blvd, Marcelino Garcia Barragan, Col. Olímpica, Guadalajara 44430, Jalisco, Mexico; blanca.aguilar@academicos.udg.mx

**Keywords:** cricket proteins, ultrasound, irradiation, enzymatic hydrolysis, fermentation, digestibility

## Abstract

The aim of this study was to evaluate the effects of γ-irradiation (IR), ultrasound (US), and combined treatments of ultrasound followed by γ-irradiation (US-IR), ultrasound followed by enzymatic hydrolysis with and without centrifugation (US-E and US-EWC, respectively), and ultrasound followed by γ-irradiation and enzymatic hydrolysis (US-IRE), on the digestibility and the nutritional value of fermented beverages containing probiotics. Results showed that US (20 min), IR (3 kGy) and US-IR (t_US_ = 20 min, dose = 3 kGy) treatments raised protein solubility from 11.5 to 21.5, 24.3 and 29.9%, respectively. According to our results, these treatments were accompanied by the increased amount of total sulfhydryl groups, surface hydrophobicity and changes to the secondary structure of the proteins measured by Fourier-transform infrared spectroscopy (FTIR). Fermented probiotic beverages, non-enriched (C) and enriched with untreated (Cr) or treated cricket protein with combined treatments were also evaluated for their in vitro protein digestibility. Results showed that the soluble fraction of US-IRE fermented beverage had the highest digestibility (94%) as compared to the whole fermented tested beverages. The peptides profile demonstrated that US-IRE had a low proportion of high molecular weight (M_W_) peptides (0.7%) and the highest proportion of low M_W_ peptides by over 80% as compared to the other treatments.

## 1. Introduction

The concern of consumers is turning towards functional foods, particularly probiotic-based products, food enriched with proteins, and polyphenols, which are becoming more and more successful and present today within a market in continuous progress. Among these functional foods are fermented dairy products with probiotic bacteria. Probiotic bacteria may prevent gastrointestinal or urogenital infections, modulate the immune system, lower cholesterol to reduce the risk of cardiovascular diseases, control blood pressure and possess anti-carcinogenic ability [1,2,3,4,5]. The stakeholders of these products always seek to improve the quality and make them more functional in order to meet the requirements of consumers and to provide healthy, balanced, natural products. Currently, much research focuses on the development of enriched products with different protein sources to provide foods with high nutritional and functional properties and to counter protein deficiency, especially for kids and elderly people. Indeed, protein-enriched products could help the elderly to prevent chronic diseases and support the most vulnerable ones suffering from immunodeficiency, bone and muscle loss [6,7]. Protein-enriched products could thus improve quality of life and contribute to the establishment of a well-balanced diet especially considering the uptake of the essential amino acids [7]. Furthermore, a particular interest in enriching dairy products with proteins has been proven due to the relationship between proteins and ferments [8]. Indeed, the incorporation of proteins can decrease the fermentation time of the dairy products, inhibit post-fermentation acidification, increase the probiotic counts during the initial stage of fermentation and delay the decline of probiotic counts during storage [9]. However, with an increasing population and the high demand for nutrients, especially proteins, finding a sustainable alternative has become a great challenge and an urgent need. Indeed, Food and Agriculture Organization (FAO) experts [10] estimated that annual food production should rise from 8.4 billion tons to nearly 13.5 billion tons to be able to feed 9 billion people in 2050. In this perspective of using alternative protein resources, algae, legumes (soybeans, peas, etc.), and insects are proposed [11]. Insects represent a valuable protein resource compared to other conventional food resources such as beef or pork [10,12,13,14], with a content of protein ranging from 35% to 63% and a high content of essential amino acids [15]. However, insect proteins have a low digestibility, due to the presence of chitin which offers some rigidity to proteins and make them resistant to the hydrolysis by digestive enzymes. Hence, insoluble precipitates can be formed, which reduces the bioavailability of trace minerals and decreases the digestibility of proteins in the small intestine [16]. In addition, the presence of high levels of hydrophobic amino acids offers a low solubility and limits the use of insect proteins in food applications. To enhance the nutritional properties and the digestibility of the proteins, several techniques could be tested. Among them, gamma irradiation (γ-irradiation) is proven to improve the nutritive value and the functional properties of proteins [17,18,19]. In addition, γ-irradiation was recognized as a safe method with a low energy cost [20,21]. Ultrasound treatment (sonication) has also been used successfully to improve the efficiency of the enzymatic hydrolysis treatment and the functional properties of proteins [22]. These techniques have the ability to alter the protein structure, resulting in the improvement of the functionality properties and the nutritional value of the proteins, adding to the benefits of a better energy efficiency by the production of peptides with high nutritional value, resulting in a good protein digestibility [23,24], food safety and nutrient preservation [25,26]. Biological processes have also demonstrated their ability to facilitate protein digestibility and it is highly suitable for human consumption. Furthermore, protein hydrolysates obtained from enzymatic hydrolysis have better functional properties such as higher solubility, absorption, good gelling and foaming properties, and consequently, higher health effects [27]. In addition, fermentation is considered as a process that decreases the levels of anti-nutrients in food grains, increasing the digestibility and their nutritive value [28,29,30,31]. Recently, combined treatments such as ultrasound pre-treatment assisted by enzymatic hydrolysis or γ-irradiation, has attracted increasing attention [32]. These combined treatments can be applied to modify the protein structure and properties through hydrolysis of covalent bonds and an increase in free sulfhydryl groups. This could contribute to an increased solubility and hydrolytic sites of the substrate, making them more accessible to the enzyme [33,34] while avoiding the restrictions linked to the rise in energy and temperature and the need for a long treatment duration. It should also be noted that the use of combined treatment of ultrasound-assisted γ-irradiation and ultrasound-assisted enzymatic hydrolysis for the treatment of proteins from insects has never been studied. Furthermore, few studies were evaluated for their effect on the nutritional value and on the digestibility of protein in general.

This study aimed to evaluate the ability of combination treatments involving sonication, γ-irradiation and pre-digestion of the protein sources with enzymes to improve the nutritional value and the functional properties of a fermented beverage with probiotics enriched with cricket proteins. Three treatments on the cricket proteins were evaluated as follows: (i) γ-irradiation assisted by sonication, (ii) enzymatic hydrolysis with alcalase assisted by sonication, and (iii) enzymatic hydrolysis assisted by sonication and γ-irradiation as a pre-treatment on the digestibility and on the nutritive value of proteins. In this context, the optimal parameters of sonication, irradiation, enzymatic hydrolysis and their combined treatments were first studied by testing their effects on the physicochemical properties of proteins (solubility, surface hydrophobicity), their structure (molecular interactions, secondary structure, content of sulfhydryl groups, peptide profiles) and in vitro digestibility via the analyses of fermented beverages vs. non-fermented counterparts.

## 2. Materials and Methods

### 2.1. Materials

The fermented and the non-fermented beverages used in this study were produced by Bio-K Plus International Inc., a Kerry company (Laval, QC, Canada). Organic cricket flour (60% protein content) was produced by Nexxus Foods (Montreal, QC, Canada). Pepsin (lyophilized powder from porcine gastric mucosa ≥ 3200 units/mg protein), trypsin (from bovine pancreas 10,000 BAEE units/mg protein) and alcalase enzyme (from *Bacillus licheniformis* ≥ 2.4 Units/g of protein) and Ellman’s reagent 5.50-dithiobis (2-nitrobenzoic acid (DTNB)) were supplied by Sigma-Aldrich (Oakville, ON, Canada). All other reagents were of analytical grade. 1-anilino-8-naphthalene sulfonate (ANS), Sodium Hydroxide (NaOH), Sulfuric Acid (H_2_SO_4_), Hydrochloric acid (HCl) and Kjeltabs Cu-1.5 were provided by Thermo Fisher Scientific (Saint-Laurent, QC, Canada).

### 2.2. Ultrasound Pre-Treatment

Ultrasound pre-treatment (US) was done using a sonicator, QSonica Q500 (model FB-505; Thermo Fisher Scientific, Ottawa, ON, Canada). US probe (model CL-334) was plunged into a flask, containing cricket powder diluted in distilled water to 40% (*w*/*v*) and the operation was conducted in a batch mode. The sonicator operated at a maximal power of 500 W and frequency of 20 kHz. Samples were treated for 10, 20, 30 and 40 min (pulsed mode: on-time 5 s and off-time 2 s), the amplitude was fixed at 70%. The flask containing the sample was immersed into a cooling bath to avoid heating induced by the US treatment. Ice-water bath was stirred every 10 min during treatment, then the suspension was freeze-dried (Labconco Freezone^®^ 2.5 L, model 7670521, Thermo Fisher Scientific) and stored in polyethylene bags prior to analyses.

### 2.3. γ-Irradiation Treatment

Irradiation treatment (IR) was applied on an aqueous suspension of cricket powder (50%) using a Cobalt 60 Underwater Calibrator UC-15A (Nordion Inc., Kanata, ON, Canada), under a dose rate of 7.7 kGy h^−1^ and at room temperature and then kept at 4 °C. The samples were treated at doses of 3, 5 and 7 kGy. Untreated samples (0 kGy) served as a control and were prepared in the same way as described previously but without irradiation. After irradiation, the solutions were freeze-dried and then stored in polyethylene bags at room temperature until used for analysis.

The combination of the treatments was applied in this study. It included ultrasound (US) procedure for 20 min as a first step for cricket protein treatment and gamma irradiation (IR) at a dose of 3 kGy as the second step. These parameters have been selected based on the results of protein solubility and structure modifications. The whole suspension of US-IR treatment was then freeze-dried and stored prior to characterization and beverage enrichment.

### 2.4. Enzymatic Hydrolysis and Combined Treatments

Enzymatic hydrolysis was realized using alcalase enzyme according to Sousa et al. [35] with modifications. The enzyme/substrate (E:S) ratio was fixed at 1:10 (*w*/*w*) and the reaction was done under agitation 100 rpm at 55 °C, pH 8.0, for 180 min. At the end of the reaction, the mixture was heated at 95 °C for 10 min to inactivate the enzyme then cooled down quickly to room temperature in an ice bath, followed by centrifugation at 13,000× *g* for 20 min [36]. The supernatant was collected and freeze-dried [37]. The whole hydrolysate (without centrifugation) was used.

The combination of ultrasound (US), γ-irradiation (IR) and enzymatic hydrolysis (E) was also performed. Three (3) combinations were carried out: ultrasound followed by enzymatic hydrolysis with and without recovery of the whole hydrolysis product (US-E and US-EWC, respectively) and ultrasound followed by γ-irradiation and enzymatic hydrolysis (US-IRE). Conditions were optimized as follows: US pre-treatment, 15 min, power of 500 W, frequency of 20 kHz and amplitude of 60% followed by γ-irradiation at 3 kGy and hydrolysis treatment as described above (alcalase enzyme for 180 min at 55 °C, pH 8.0). The products resulting from these combinations were used for the enrichment of the probiotic-based beverage.

### 2.5. Beverage Preparation

A commercial Bio-K+ Blueberry fermented beverage containing *L. acidophilus* CL1285, *L. casei* LBC80R and *L. rhamnosus* CLR2 as probiotics was used for enrichment. The beverage has been enriched with cricket powder having a total protein content of 13% and the proteins were pre-treated with selected processes as described above. Other beverages non-enriched (Bio-K+ Blueberry, with 3% of protein) and non-fermented were produced and were used as controls for comparison.

### 2.6. Protein Solubility

Protein solubility was determined according to a method of Adebiyi and Aluko [38]. A quantity of 100 mg of total proteins was dissolved in distilled water and then vortexed for 60 min at 25 °C. Samples were centrifuged for 15 min at 15,000× *g*. The protein content of the supernatant was determined by the Kjeldahl method using a Foss Kjeldahl system (Foss, Eden Prairie, MN, USA) with an automation consisting of 6 steps: (i) test portion and reagent addition, (ii) initial and final digestion, (iii) cooling and dilution, (iv) NaOH addition, (v) steam distillation and titration, (vi) automatic pumping of flask contents to waste. A volume of 4–6 mL of test portion was poured into a 250 mL Foss digestion tube. A number of 2 catalyst copper Foss Kjeltabs Cu-1.5 were added and 15 mL of concentrated H_2_SO_4_ was carefully poured into the tube. Digestion was performed at 420 °C for 2.5 h by using a Digestor Labtec™ Line DT208 (Foss). After cooling down, samples were diluted with 80 mL distilled water and distillation was performed using a Distillation Unit Kjeltec™ 8200. 

Then, distillates were titrated with 0.1 M HCl and results were expressed in percent proteins (g/100 g), following calculations using AOAC 991.20 method [39], analysis and quality evaluation of milk and milk products. The protein content was obtained by using the following Equations (1) and (2):% Nitrogen = [(V_sample_ (mL) − V_blank_ (mL)) × N × 14.007) / wt test portion (mg)] × 100(1)
where V_sample_ is the volume of titrant used for titrating the sample (mL), V_blank_ is the volume of titrant used for titrating the blank (mL), N is the normality of titrant (standard HCl), 14.007 is the atomic weight of nitrogen element, and wt test portion is the weight (mg) of test portion.
% Protein = % Nitrogen × 6.25(2)
where 6.25 is the protein factor.

Subsequently, the percentage of soluble proteins was calculated according to Equation (3):% solubility = Quantity of total proteins in the supernatant/Quantity of total proteins in the sample * 100(3)

### 2.7. Measurement of Sulfhydryl (SH) Bonds

The total sulfhydryl bonds (SH) content of cricket proteins was determined using Ellman’s reagent 5.50-dithiobis (2-nitrobenzoic acid/DTNB) and according to Ellman’s procedure 1959 with some modification [40]. Samples were prepared into 1% suspensions of phosphate buffer (86 mM, pH 7.0). A volume of 0.5 mL of sample suspension was then mixed with 5 mL of urea buffer (pH 8, dissolved 10.4 g of Tris, 1.2 g of EDTA, 6.9 g of glycine, and 480 g of urea in deionized Milli-Q water to 1 L) and 20 µL of 4 mg/mL DTNB then incubated for 30 min at room temperature. Absorbance was read at 412 nm using a UV-Vis spectrophotometer Cary 8454 (Agilent technologies, Mississauga, ON, Canada). The surface free SH groups were determined using the same method but without urea. The total SH group content was calculated as follows:C_SH_ = 73.53 A_SH_/C_S_(4)
where C_SH_ is the content of sulfhydryl groups (µmol/g), A_SH_ is the absorbance at 412 nm, C_S_ is the sample concentration (mg/mL).

### 2.8. Surface Hydrophobicity

The surface hydrophobicity of cricket proteins was determined according to a method described by Kato and Nakai [41], using an ANS probe. A series of sample dilutions were prepared at 0.00125, 0.0025, 0.005, 0.01, 0.02% (*w*/*w*) with 0.01 M Na_2_HPO_4_ buffer (0.1 M, pH 7). A quantity of 20 µL of ANS (1-anilino-8-naphthalene sulfonate) solution (8 mM dissolved in 0.01 M phosphate buffer pH 7) was added into 4 mL diluted protein solution, shaken immediately and then the fluorescence intensity was measured at 25 °C with a Tecan Infinite M1000 Pro fluorescence microplate reader (Tecan Austria GmbH, Austria) set up to an excitation wavelength of 390 nm and an emission wavelength of 468 nm. Surface hydrophobicity (Ho) was expressed as the initial slope of the plot of fluorescence intensity versus protein concentration (mg/mL) plot.

### 2.9. Molecular Characterization by Fourier Transform Infrared (FTIR) Spectroscopy

The FTIR analyses of treated cricket proteins were carried out by using an ATR-FTIR Spectrum One spectrometer (PerkinElmer, Woodbridge, ON, Canada), equipped with an attenuated total reflectance device for solids analysis and a high linearity lithium tantalate detector. Lyophilized samples treated by ultrasound (US), γ-irradiation (IR), and ultrasound-assisted γ-irradiation (US-IR) were subjected to analyses in comparison with the untreated samples (control). They were analyzed at room temperature in the range of 4000 to 650 cm^−1^ using an attenuated total reflectance (ATR) mode of operation. A number of 64 scans were accumulated at a resolution of 4 cm^−1^. After attenuation of total reflectance and baseline correction, spectra were normalized with a limit ordinate of 1.5 absorbance units. The secondary structure of proteins was also studied within the Amide I region (1700–1600 cm^−1^) using the method of second derivative spectra according to Byler and Susi [42]. The different Amide I components (α-helix, β-sheet, β-turn, unfolded/random coils) were identified and conformation changes were estimated by calculating sample/control ratios, based on the intensity of the negative bands of derivative spectra.

### 2.10. In Vitro Digestibility

The in vitro digestibility of fermented and non-fermented beverages, non-enriched and enriched with treated and non-treated cricket proteins, was performed according to Wang et al. [43] with minor modifications. Sample digestion test was carried out using pepsin and trypsin sequential digestion treatment. A quantity of 0.5 g of protein sample was dissolved in 9.5 mL of 0.1 M HCl and digested at 37 °C with 10 mg of pepsin previously dissolved in 0.5 mL of 0.1 M HCl at 1:50 (*w*/*v*) E:S ratio and a stirring speed of 100 rpm. Sample aliquots (2 mL) were taken at 0, 30, 60, 90 and 120 min, respectively. After pepsin digestion, the mixture was adjusted to pH 8.0 using 1 M NaOH. Then, 10 mg of trypsin was added at a 1:50 E:S ratio, at 37 °C and 100 rpm. Samples of 2 mL were taken at various digestion times during an additional period of 120 min. The soluble nitrogen release during the digestion process was determined using the trichloroacetic acid (TCA) precipitation method at a final concentration of 10% [37]. For each aliquot taken during pepsin and trypsin digestion, a volume of 2 mL of 20% TCA was added to the pepsin and/or trypsin hydrolysates. The precipitate was then centrifuged at 10,000× *g* for 20 min, and the supernatant was assayed for nitrogen content using the Kjeldahl method as described above. The percentage of nitrogen released was defined as follows:% Nitrogen release = (Nitrogen content in the supernatant/Nitrogen content in the sample) × 100(5)

### 2.11. Peptide Profile (SEC-HPLC)

A volume of 1 mL of each sample was taken at various time of digestion, heated at 95 °C for 5 min to inactivate the digestive enzymes, centrifuged at 10,000× *g* for 20 min and the supernatant was then recovered and filtered through a 0.2 μm filter and analyzed by SEC-HPLC. The molecular weight distribution of soluble fractions of peptides in fermented and non-fermented beverages was carried out using an Agilent 1260 Infinity HPLC System (Agilent Technologies Canada Inc., Mississauga, ON, Canada) equipped with a size-exclusion column Biosep 5 µm SEC-s2000 145 Å (particle size 5 µm, pore size 145 Å, 300 × 7.8 mm; Phenomenex Inc., Torrance, CA, USA) and a guard column (SecurityGuard cartridge for GFC 2000; 4 × 3 mm; Phenomenex Inc.). A 100 mM sodium phosphate buffer solution (pH 6.8) was used as the mobile phase. The volume of sample was 20 μL and the analysis was carried out at ambient temperature using a flow rate of 1 mL/min for 20 min. Detection was performed at 280 nm using a 1260 diode array detector (DAD). The peptides were identified by comparing the retention times with standard proteins such as bovine thyroglobulin (670 kDa), IgA (300 kDa), IgG (150 kDa), ovalbumin (44 kDa), myoglobin (17 kDa) and uridine (244.2 Da). Chromatograms were collected by using OpenLAB ChemStation software (Rev.C.01.07 SR2 [255], Beijing, China). The baseline was corrected manually, and the total surface area of chromatograms was integrated and split into 3 groups of molecular weights (>3000 Da; 3000-260 Da; <260 Da) using the function split manually integrated peaks and expressed as a percentage of the total area. Analysis of peptide profile was performed on fermented and non-fermented control, Cr, US-IR, US-E, US-EWC and US-IRE beverages before and during in vitro digestion.

The relative difference of M_W_ distribution between the end and the beginning of the in vitro digestion (Δ value) was determined as follows:Δ = %E−%B/%F(6)

### 2.12. Statistical Analysis

All treatments and analyses were performed in triplicate (*n* = 3). The results were reported as mean values ± standard deviation. Statistical analysis was carried out using SPSS software version 22 (IBM Corporation, Somers, NY, USA) and the significant differences (*p* ≤ 0.05) between the means were evaluated using one-way analysis of variance (ANOVA). Duncan’s multiple range tests for equal variances and Tamhane’s test for unequal variances were performed for statistical analysis.

## 3. Results and Discussion

### 3.1. Protein Solubility

Measurement of protein solubility was used to characterize the fragmentation or agglomeration of cricket proteins before and after all treatments (US, IR and US-IR). The results of the effect of US are presented in Figure 1a and show that the US treatment significantly increased the protein solubility (*p* ≤ 0.05) as compared to the untreated samples. However, an increase of US time (from 10 to 40 min) did not increase the solubility (*p* > 0.05), meaning that a time of 10 min was sufficient to reach a high solubility of proteins (>20%). This effect could be due to a linkage break between chitin and proteins, resulting from a modification of electrostatic links, which facilitates the redirection of hydrophilic amino acids to the aqueous phase [44,45]. Other observations explain that an increase of protein solubility could be due to the capacity of US treatment to lead changes in the three-dimensional structure of proteins, associated with interactions between the hydrophobic and hydrophilic surface, and resulting in higher electrostatic forces that increase protein–water interactions, and hence solubility [46,47,48].

The effect of γ-irradiation treatment is shown in Figure 1b. A significant increase (*p* ≤ 0.05) in protein solubility was observed at an optimal dose of 3 kGy compared to untreated samples and other treatments. Indeed, the protein solubility increased from 11.6 to 24.4% (~2-fold) after irradiation treatment of 3 kGy. Similarly, Hassan et al. [49] found that doses higher than 0.5 kGy significantly enhanced the protein solubility for sesame protein with an optimal dose of 1 kGy. In addition, Dogbevi et al. [50] reported that the increase of protein solubility was due to the unfolding of proteins through their deamination during irradiation, which can change the protein statutes from water anti-binding to water binding by the conversion of amide groups into acid groups [51]. In the current study, the increase in protein solubility could be attributed to the destruction of chitin-protein binding by the application of γ-irradiation which further favors the deamination of protein molecules [52]. On the other hand, the decrease of protein solubility observed at doses > 3 kGy could be explained by their denaturation due to the aggregation of unfolded proteins to form a high molecular weight protein network [53]. In addition, Maity et al. [54] reported a decrease in the solubility of vegetable proteins from *Oryza sativa L.* after a treatment of 6 kGy. The combined treatment US-IR (Figure 1c) was realized using the irradiation dose (3 kGy) and ultrasound duration (t_US_ = 20 min). Results show that this treatment was able to improve the protein solubility from 11.6 to 29.9% as compared to the untreated sample. The energy delivered by the combination of ultrasound and γ-irradiation has further improved the fragmentation of proteins and consequently led to more soluble proteins following the degradation of small protein aggregates.

### 3.2. Sulfhydryl Groups and Surface Hydrophobicity

The effect of US treatment at increasing durations (0, 10, 20, 30 and 40 min) on the content of total free sulfhydryl (SH) groups, surface free SH groups and on the surface hydrophobicity of cricket proteins is presented in Figure 2a. The results of the content of surface and total SH groups significantly increased (*p* ≤ 0.05) after the application of US treatment. However, no significant difference (*p* > 0.05) of total SH groups was observed between all treatments with a range of 2.48–3.22 μmol/g. Only slight differences (*p* ≤ 0.05) of surface SH groups were observed between treatments, with a maximum of 0.54 μmol/g obtained after 20 min of treatment. The surface hydrophobicity (Ho) was also significantly increased after treatments (*p* ≤ 0.05), with maximum values observed after 20 min and 30 min showing a hydrophobicity of 5210 and 5844, respectively, and no significant difference between these two treatments was observed (*p* > 0.05). These observations are probably due to the unfolding of proteins, following the application of ultrasound, and may have led to the exposure of hydrophobic amino acids or clusters and the SH-containing amino groups to the external surface of protein molecules [55,56]. Similar results have been reported by Higuera-Barraza et al. [57], who applied ultrasound treatment at different times in squid (*Dosidicus gigas*) mantle proteins, finding a significant increase in Ho according to the time and amplitude used compared to the untreated samples. The effect of IR treatment on SH groups and surface hydrophobicity is shown in Figure 2b. The content of free SH increased with increasing irradiation doses, from 0.69 to 3.79 μmol/g at 7 kGy, the maximum irradiation conditions for the release of SH groups. In addition, the maximum content of surface SH was obtained at 3 kGy, showing a significantly higher value of 0.64 μmol/g (*p* ≤ 0.05) compared to 0.22 μmol/g for untreated proteins. The surface hydrophobicity increased significantly from 2601 to 5097 after a treatment of 5 kGy and then decreased to a value of 3881 at 7 kGy. No significant difference (*p* > 0.05) was observed between the treatments of 3 kGy (4394.6) and 5 kGy (5097) which are considered as optimal treatments to increase the Ho of proteins. This observation is in contradiction with the results obtained for proteins solubility, which showed a decrease in the solubility with an increase in the irradiation dose. We suggest that a high irradiation dose might have the potential to break disulfide and hydrogen bonds, ionic and hydrophobic interactions and lead to an increase in total SH content [58]. The decrease in solubility with the dose of 7 kGy could be associated with a Maillard reaction, when the amino groups of lysine react with the carbonyl groups of reducing sugars, as reported by Hooshmand and Klopfenstein [59] after irradiation of maize and wheat flours at 7.5 kGy. It could also be explained by a high irradiation dose effect on non disulfide covalent bonds that form and cause the closing of protein structures. Consequently, some peptide bonds were masked or smaller peptides polymerized to form a highly cross-linked protein network [58]. The combined treatment US-IR (Figure 2c) generated a significant increase in SH content (*p* ≤ 0.05) compared to the untreated sample by reaching 2.87 μmol/g, but no significant difference (*p* > 0.05) was noted compared to US alone (2.67 μmol/g) and IR alone (2.58 μmol/g). In counterpart, US-IR treatment showed the highest content of surface SH (1.13 μmol/g), significantly higher (*p* ≤ 0.05) than control (0.22 μmol/g) and individual treatments (0.54 μmol/g for US and 0.64 μmol/g for IR). Similarly, the surface hydrophobicity increased significantly (*p* ≤ 0.05) with the combined treatment compared to the untreated and IR samples, but no significant difference (*p* > 0.05) was observed between US-IR and US alone treated samples. From these observations, although US-IR treatment contributed to a significant increase of protein solubility, it did not trigger a major change in total SH bonds compared to individual treatments. However, US-IR treatment induced a significantly higher content of surface SH groups (*p* ≤ 0.05) compared to individual treatments, and a higher Ho (*p* ≤ 0.05) compared to IR alone without affecting the Ho level of proteins treated by US alone. A possible explanation is that the combined treatment led to protein unfolding, thereby exposing SH groups to the outer surface of the proteins, which may have a functional or structural role in proteins. Overall, these trends observed in Figure 2 are in accordance with solubility results (Figure 1), as protein solubility increased with the SH content and the Ho of treated proteins [60]. 

### 3.3. ATR-FTIR Analysis

ATR-FTIR spectroscopy of cricket proteins was performed to verify the effect of US, IR and combined US-IR treatments on the intermolecular interactions associated with the presumed hydrolysis of cricket proteins. Primary spectra of samples treated by the different treatments are presented in Figure 3. The results show that the intensities of some bands were affected by the type of treatment. In the quantitative region (3600–3000 cm^−1^), it is clear that the O-H and N-H stretching vibrations—comprising Amide A band at 3300 cm^−1^ (O-H stretching) and Amide B band at 3100 cm^−1^ (N-H stretching)—increased after treatments, mainly due to the presence of more free amino and hydroxyl groups, and most likely related to partial hydrolysis. However, no evidence of change in hydrogen bonding was systematically observed between all treated samples and control as no significant shift of the O-H stretching mode to lower frequencies was observed, most likely due to the fact these doses did not influence the hydrogen bonding of solid proteins at the different doses of US and IR. Furthermore, the bands related to the secondary structure of proteins also increased in intensity after treatment compared to the untreated samples. Indeed, Figure 3 also shows that the Amide I band at 1628 cm^−1^ (C=O stretching coupled to C-N stretching and N-H bending) and the Amide II band at 1535 cm^−1^ (referred to C-N stretching coupled to N-H bending), generally showed higher intensities after treatments of proteins. These structural changes of protein configuration have a noticeable influence on the protein stability, as previously reported by Vargas et al. [61]. Within the region 1700–1600 cm^−1^ (Amide I), the main band has been identified as being sensitive to particular secondary structural changes, as previously reported [62,63,64]. Therefore, the estimated secondary structure of cricket proteins subjected to the different treatments was determined in this infrared region and listed in Table 1.

Regarding US treatment (Figure 3a), the highest intensities in typical peaks related to proteins (Amide A, B, I and II) correspond to the treatments of 20 and 30 min whereas lower intensities were observed at 10 and 40 min. The increase of the intensity in Amide I and Amide II bands observed at 20 and 30 min may be due to the fragmentation of cricket proteins that could lead to the unfolding of the compact protein structure and which is possibly attributed to more intensive interactions between water and protein [26]. Similarly, the effect of the US duration on the proteins structure was evaluated by other authors. Hence, it was found that the application of US treatment on β-lactoglobulin [65] and on chicken bones [36] could lead to secondary structural changes, suggesting that US treatment could disrupt intermolecular interactions and decrease hydrogen bonding, resulting in more peptide release. Results of Table 1 indicate an increase mainly in α-helix, β-sheet, random coil and β-turn structures of cricket proteins, and these changes were altogether observed at 20 min. The increase in α-helix and β-sheet under the effect of US treatment was also reported by Chandrapala et al. [66] and Ma et al. [67]. These spectral changes reflect the disruption on the interactions between different parts of the protein molecules induced by US treatment which led to conformation changes [68]. It is to be noted that, although a lower absorbance of the Amide I band was observed at 40 min of US treatment compared to 20 and 30 min, the second derivative function indicates no significant variation of α-helix (ratio treatment/control of 0.96 at 1653 cm^−1^) compared to a higher level of β-sheets and β-turns, hence suggesting a highly stabilized protein network at this dose. This observation coincides with the results of surface hydrophobicity analysis which showed a higher Ho after 20 min of the ultrasound treatment. This suggests that US treatment for 20 min may result in conformational changes relative to a more stable, ordered structure.

Figure 3b shows the FTIR profile of samples subjected to IR treatment. Results show an increase in the peak intensity of Amide A and Amide B bands but no change in the Amide I mode, after application of γ-irradiation. Moreover, it is interesting to note that the absorbances of Amide A and B decrease with the irradiation dose (from 3 to 7 kGy). This result suggests that γ-irradiation changed the protein structure in such a way that was attributed to the release of peptides due to protein degradation. Similar observations were reported by Maity [54] for sunflower protein isolate. The influence of irradiation on the secondary structure of cricket proteins is shown in Table 1. The results indicate that, compared to the control, an important decrease of α-helix structure—and to a lesser degree of β-sheets—was observed towards a substantial increment of random coils and β-turns (at 1663 cm^−1^) in irradiated samples. These changes are particularly observed at 3 kGy. Thus, it can be inferred that under γ-irradiation proteins might unfold with weakened intermolecular hydrogen bonds, causing the disruption of β-sheets and favoring the formation of unordered structures. It was observed by Malik et al. [69] that γ-irradiation decreased α-helix content and increased the β-sheets, but also the β-turn and random coil contents of sunflower protein isolate. Lee and Song [70] explained the change in the secondary structure of myoglobin by the cleavage of covalent bonds and the formation of aggregated products. Similar results were found by Le Tien et al. [71] who observed a decrease of α-helical structure content of whey protein-based films due to the ability of γ-irradiation to alter protein conformation.

Finally, as shown in Figure 3c, cricket proteins treated with US-IR combined process showed the highest intensity of –OH and –NH stretching bonds, but also high intensities of Amide I and Amide II bonds (intermediate between single treatments) compared to the untreated and treated samples with US (20 min) and IR (3 kGy) alone. This observation is in accordance with the important protein solubility of samples treated by US-IR (Figure 1c), suggesting a potential higher degree of hydrolysis or a higher exposure of hydroxyl and amino groups of the proteins to solvation after the combined treatment. Moreover, the combined treatment affected more significantly the random coil secondary structure of proteins, as shown in Table 1 Indeed, US-IR treatment generally presented a similar effect to that of IR single treatment at 3 kGy, but a synergic effect on random coils was measured with a high ratio of 3.47 compared to untreated proteins. Thus, US-IR treatment could form a synergistic potentialization for structure modification of proteins through the destruction of covalent bonds, molecular interactions or conformational ordering. Therefore, these results suggest that infrared absorbances of primary as well as derivative spectra of treated proteins were greatly affected by the local environment of the different structural groups related to protein conformation. Sonication and irradiation caused microstructural changes at the molecular level. These changes may be attributed to fragmentation, cross-linking and aggregation, as previously observed by other authors [61,72,73,74,75].

### 3.4. In Vitro Digestibility

Different effects of US, IR and enzymatic hydrolysis of the cricket proteins on the protein digestibility in the enriched beverages were evaluated and results are presented in Figure 4. The in vitro digestibility test was monitored during 2 h of pepsin treatment followed by 2 h of trypsin treatment. As illustrated in Figure 4a, the percentage of digestibility (nitrogen release) of non-fermented (C NF) and fermented (C F) non-enriched beverage (control) is presented all along the in vitro digestion time. Figure 4b presents the effect of US-IR combined treatment, i.e., the digestibility of non-fermented vs. fermented enriched beverages containing cricket proteins (Cr NF and Cr F, respectively), and non-fermented vs. fermented beverages containing cricket proteins treated by US-IR process (US-IR NF and US-IR F, respectively). Figure 4c illustrates the effect of an additional enzymatic treatment, i.e., the digestibility of non-fermented vs. fermented beverages containing cricket proteins: (i) treated by combined US and enzymatic hydrolysis, the soluble part (US-E NF and US-E F, respectively), (ii) treated by combined US and enzymatic hydrolysis using the whole product of hydrolysis (US-EWC NF and US-EWC F), and (iii) treated by combined US, IR and enzymatic hydrolysis (US-IRE NF and U-IRE F, respectively).

Results show that the nitrogen release increased during the digestion in all samples, but the profile of nitrogen release is different depending the treatment applied on proteins. In addition, an improved digestibility was observed in all fermented samples (assignation F) as compared to their respective non-fermented counterparts. In particular, the digestibility of enriched beverages was reached more rapidly in products containing treated cricket proteins (Figure 4b,c) compared to non-enriched ones (Figure 4a). Indeed, the region of 30–40% nitrogen was reached at 120 min for non-enriched control samples (Figure 4a) and measurements evolved linearly up to the region of 50–55% nitrogen at 240 min, with very similar curves in the portion of 120–240 min. In comparison, for enriched beverages with or without US-IR treatment (Figure 4b), the region of 30–40% nitrogen was reached after only 30 min and a plateau at 45% nitrogen was observed in untreated samples after 120 min, whereas a continuous slight increase was measured in treated samples (US-IR NF and US-IS F) up to 55% of nitrogen at 120 min. This improvement of nitrogen release was expectedly more noticeable in beverages containing cricket proteins treated with enzymatic hydrolysis (Figure 4c)—with much higher percentages at 0 min comprised between 40 and 90%, compared to 10–15% in other treatments presented in Figure 4a,b—and the higher digestion of fermented products (F) was evidenced compared to non-fermented (NF) ones. Indeed, non-fermented and fermented samples submitted to combined US and enzymatic hydrolysis before centrifugation (US-EWC NF and US-EWC F) showed the lowest percentages of nitrogen throughout all digestion. Although a high percentage of 40 and 47% was found at 0 min, respectively, a low final value of 57% was found after 240 min, which is equivalent to US-IR NF and US-IR F samples (without enzymatic treatment) in Figure 4b. However, the soluble part (supernatant) obtained from combined US and enzyme treatment after centrifugation (US-E NF and US-E F) induced an increase of nitrogen release during digestion, starting at 74–76% at 0 min and increasing to 81 and 94% at 240 min for US-E NF and US-E F samples, respectively. Regarding the treatment combining ultrasound, enzyme and irradiation (US-IRE NF and US-IRE F), nitrogen release started from 57 and 90% at 0 min, and a release of 66 and 94% was reached at 120 min after pepsin simulated digestion, followed by a plateau from 120 to 240 min (trypsin digestion). Overall, the most digestible products were found to be US-E F and US-IRE F beverages. These results are mainly due to the role of the enzymatic pre-treatment which ensures a pre-digestion of proteins by link breakage between proteins and other constituents, which prevent the good digestibility of cricket proteins, in particular chitin, as already reported by other studies [16,76,77]. Furthermore, the dissociated state of the proteins resulting from enzymatic hydrolysis promotes an easier digestion [78]. In addition, this finding was consistent with the result of Lacroix et al. [37], which showed that the isolation of the by-products of molecular weight lower than 5000 Da using the ultrafiltration procedure has allowed a significant increase on the digestibility and the net protein ratio of rapeseed proteins hydrolysate. In addition, considering the proteins without enzymatic treatment, the US-IR beverage is associated with improved digestibility (*p* ≤ 0.05) compared to the non-enriched control beverage (C) and the enriched beverage (Cr). Previous works found that the application of US pre-treatment ensures the degradation of protein structure, enhances peptide cleavage and improves protein digestibility [79], but also US could extract the bioactive peptides from the biological matrix [80]. Likewise, γ-irradiation has been shown to break down proteins in a random way to produce peptides and improve digestibility [81,82]. These observations are in accordance with the results obtained in this study treated with these processes and which have succeeded in an improved solubility, the release of total and surface SH bonds, and changes in the secondary structure of proteins. Furthermore, US-IR-assisted enzymatic hydrolysis (US-IRE) resulted in a faster digestion than the US-E process. This observation may be due to the structural changes and exposure of more easily digestible peptides and amino acids with digestive enzymes [83], associated with irradiation that probably promoted the digestion and utilization of hydrolysates primarily by ferments and secondly by digestive enzymes [49,58]. Thus, this proposed mechanism could explain the pronounced difference of digestibility observed between the US-IRE NF and US-IRE F beverages.

### 3.5. Molecular Weight (M_w_) Distribution by SEC Analysis

The molecular weight (M_w_) distribution of protein hydrolysates obtained during the digestion of the different NF and F beverages (Control, Cr, US-IR, US-E, US-EWC and US-IRE) is presented in Table 2. The results show that the control beverages (NF and F) had a high percentage of medium M_w_ peptides (MMW between 3000–260 Da) with values of 63–64% at the end of the digestion period, a low percentage of low M_w_ peptides (LMW < 260 Da) at 34% and a very low fraction of high Mw peptides (HMW > 3000 Da) at 2% of peptides. Conversely, enriched beverages with proteins (Cr, US-IR, US-E, US-EWC and US-IRE) were characterized by a high proportion of LMW peptides with percentages around 51–53% for Cr samples, 48–50% for US-IR, 57–58% for US-E, 65–69% for US-EWC and approximately 81% for US-IRE samples, at the end of digestion. This higher percentage of LMW peptides was observed at the expense of a significant decrease of MMW peptides, with values around 43–45% for Cr samples, 46–49% for US-IR, 41–42% for US-E, 30–35% for US-EWC and 18–19% for US-IRE samples, at the end of digestion. The percentage of HMW peptides remained very low regardless of the treatment, with higher values of 3.0–3.5% in Cr and US-IR samples compared to control, and lower values ranging from 0.3 to 0.9% under enzymatic treatments (US-E, US-EWC and US-IRE), at the end of digestion.

Otherwise, should be noted that a decrease in the percentage of HMW peptides was observed all along the pepsin-trypsin digestion in all samples, as this fraction may be more sensitive to the different treatments. Furthermore, the fermented samples US-E F, US-EWC F and US-IRE F are characterized by a very weak or negligible percentage of HMW peptides accompanied by the highest percentage of LMW peptides compared to the other treatments without enzyme (control, Cr, US-IR). These results reflect the high degree of protein degradation in these products, and some authors have suggested that the enrichment of beverages with proteins under hydrolysates form allows enhanced protein digestibility and tends to increase the level of functional and bioactive peptides [84,85]. Additionally, the pre-digestion of cricket protein with the US-E and US-IRE treatments could facilitate the role of pepsin and trypsin enzymes during the digestion but could also mask the proteolytic activity of ferments during fermentation. In the same context, Li et al. [86] found that suitable enzymatic hydrolysis was an optimal approach to improve some physical properties, immunoreactivity and in vitro protein digestibility of soy protein isolate for infant formula. The profile change over the US-IRE treatment may be explained by the proteolysis of large insoluble proteins into soluble peptides. Hence, hydrolysis into smaller peptides confirmed the fast digestion when the enzymatic hydrolysis was combined with γ-irradiation, ensuring the breakdown of polypeptide chains and a strong formation of small peptides [75].

In order to better assess the efficiency of peptide hydrolysis with and without fermentation, the relative difference of M_W_ distribution between the end and the beginning of the in vitro digestion (Δ value) was determined. The analysis of the peptide profile by SEC-HPLC did not show a critical change caused by fermentation, except for the US-IR treatment as related to MMW and LMW peptides. In general, the Δ values in other treatments were slightly or not at all influenced by the fermentation. However, for US-IR treatment, the fermentation impacted more the conversion of MMW and LMW peptides, with Δ_NF_ = 32.3% and Δ_F_ = 2.9% for MMW peptides and Δ_NF_ = −18.4% and Δ_F_ = −0.6% for LMW peptides. These data suggest that the fermentation process generated a much less sensitive product to US-IR treatment as it reduced considerably the conversion of peptides into lower Mw during digestion. This observation can be explained by the fact that for fermented products, a pre-digestion started during fermentation which conducted to a faster hydrolysis by digestive enzymes under the combined effect of ultrasound and γ-irradiation. Furthermore, this could be corroborated by the occurrence that pepsin and trypsin cleaved additional peptide bonds that were not cleaved previously by ferments. According to Ogodo et al. [87] and Pranoto et al. [88], the degradation of high M_w_ peptides into smaller ones, produced during fermentation and digestion, has a significant effect on the digestibility of proteins.

## 4. Conclusions

Industries that manufacture functional beverages are looking for alternatives to improve the nutritional quality of their products as well as their shelf life. Insect protein is a rich source of amino acids, and in this work the effect of ultrasound (US), γ-irradiation (IR), and enzymatic hydrolysis (E) of cricket proteins and their combined processes on the physicochemical, structural properties and in vitro digestibility on fermented beverages enriched with such cricket protein was analyzed. Combined treatment of ultrasound-assisted γ-irradiation (US-IR; dose = 3 kGy, t_US_ = 20 min) improved the solubility of cricket proteins and increased their surface and total sulfhydryl group content. This also ensured a higher level of hydroxyl/amino groups susceptible to solvation associated with a reduction of random coils in protein conformation possibly associated with aggregation, as shown by FTIR analysis, as well as a high level of medium and low Mw peptides after in vitro digestion, as shown by HPLC/SEC analysis. This treatment showed a high digestibility compared to the non-enriched and enriched beverages, but the treatments combined with enzymatic hydrolysis US-E and US-IRE induced a much higher positive effect in improving the digestibility of fermented (F) beverages and higher formation of low M_w_ peptides after digestion (up to 81%). Therefore, it is reasonable to assume that a combined treatment of cricket proteins is very promising to ensure a high digestibility of protein-enriched beverages with very satisfying functional and nutritional properties. The results obtained from this study offer a new, promising perspective for the valorization of cricket proteins in the application of functional foods.

## Figures and Tables

**Figure 1 foods-10-02259-f001:**
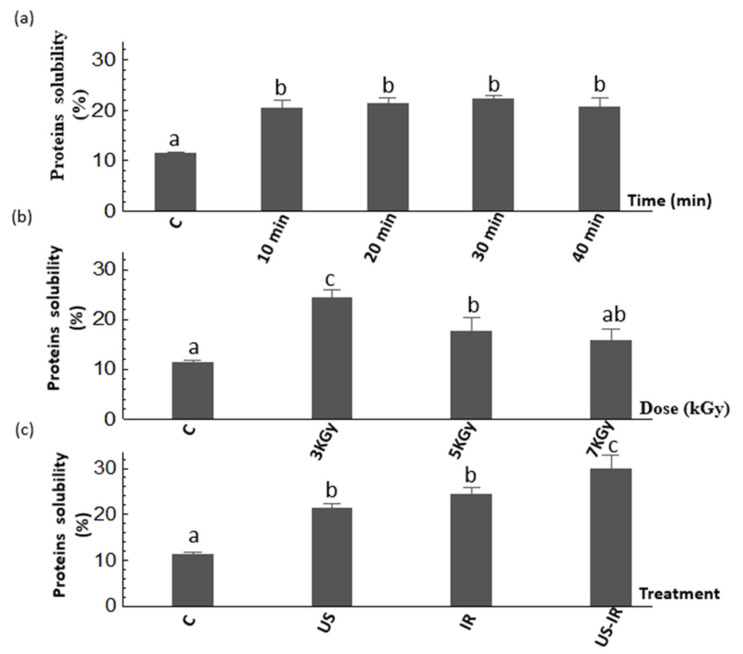
Solubility of cricket proteins in untreated samples (C), samples treated by ultrasound (US) at increasing times (10, 20, 30 and 40 min) (**a**), by γ-irradiation (IR) at increasing doses (3, 5 and 7 kGy) (**b**), and by combined ultrasound-assisted γ-irradiation (US-IR; 20 min; 3 kGy) (**c**). Different letters above the bars indicate significant differences among the mean values of the samples (*p* ≤ 0.05). Data shown is the mean ± SD (*n* = 3).

**Figure 2 foods-10-02259-f002:**
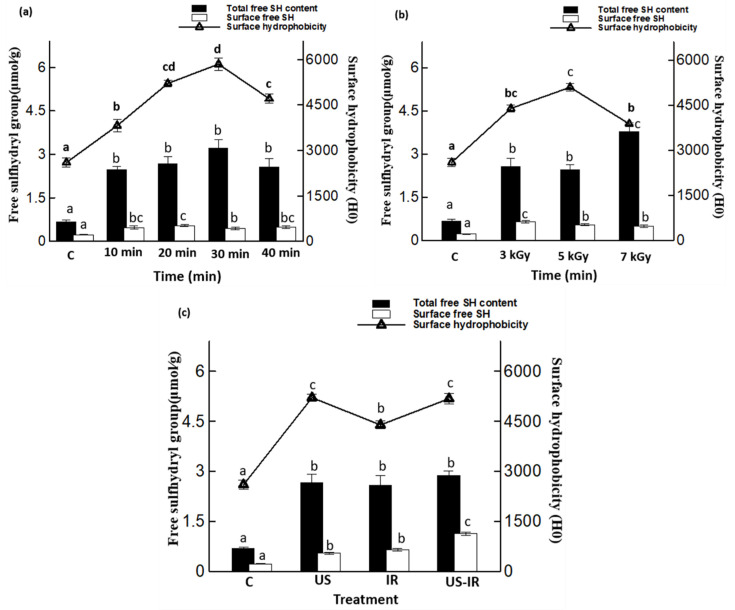
Sulfhydryl group content in cricket proteins for untreated samples (C), samples treated by ultrasound (US) at different times (10, 20, 30 and 40 min) (**a**), by γ-irradiation at different doses (3, 5 and 7 kGy) (**b**), and by combined ultrasound-assisted γ-irradiation (US-IR; 20 min; 3 kGy) (**c**). Different letters above the bars indicate significant differences among the mean values of the samples (*p* ≤ 0.05). Data shown is the mean ± SD (*n* = 3).

**Figure 3 foods-10-02259-f003:**
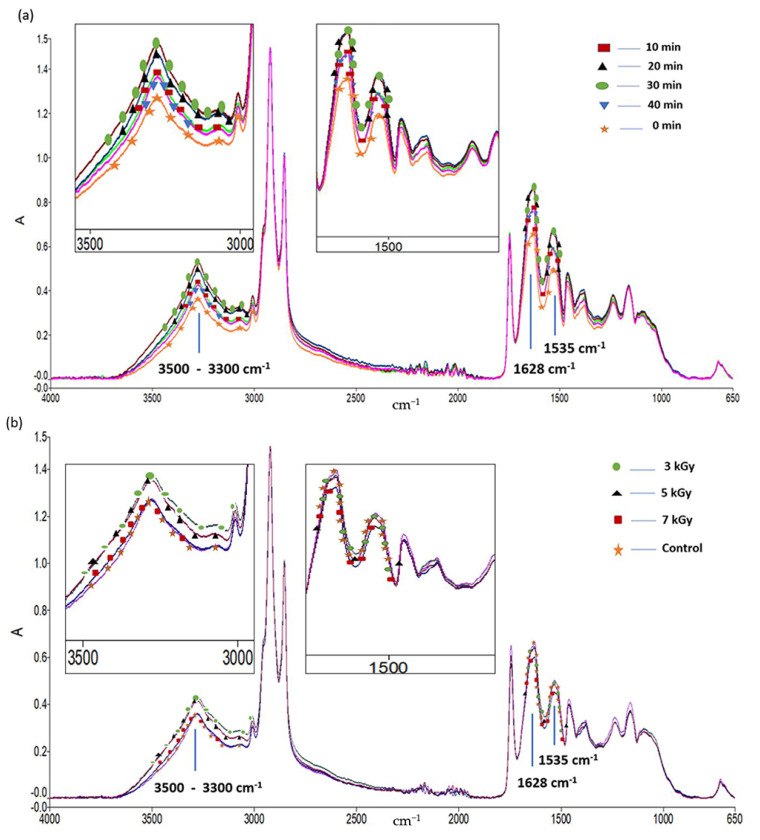
FTIR spectra of cricket proteins before treatments (control), after ultrasound (US) treatment (10, 20, 30 and 40 min) (**a**), γ -irradiation (IR) treatment (3, 5 and 7 kGy) (**b**), and after single and combined treatments US, IR and IR assisted with US (US-IR) (**c**). For each global spectrum, focused areas in the regions 3500–3300 cm^−1^ and 1628–1200 cm^−1^ indicate the main vibration changes related to the secondary structure of proteins.

**Figure 4 foods-10-02259-f004:**
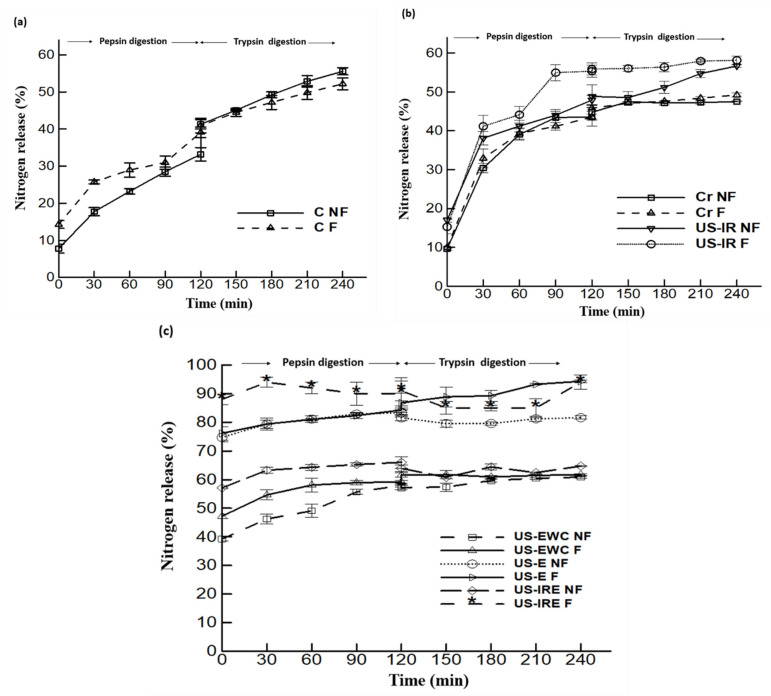
Percentage of nitrogen released from fermented (F) and non-fermented (NF) non-enriched (control C) (**a**), untreated cricket protein-enriched (Cr) beverages, treated with γ-irradiation assisted with ultrasound (US-IR) (**b**), and treated cricket protein-enriched beverages subjected to enzymatic hydrolysis assisted with ultrasound (US-E), treated cricket protein-enriched beverages subjected to enzymatic hydrolysis assisted with ultrasound the whole product of hydrolysis (US-EWC), to enzymatic hydrolysis assisted with γ-irradiation and ultrasound (US-IRE) (**c**) during pepsin (120 min) and trypsin (120 min) simulated digestion. Data shown is the mean ± SD, *n* = 3.

**Table 1 foods-10-02259-t001:** Effect of ultrasound treatment on the secondary structure of cricket protein. Data were collected from the amplitude of negative bands of second derivative spectra in the Amide I region (1700–1600 cm^−1^) and the ratio of treatment/control was calculated.

Treatment Effect (Ratio Treatment/Control)	α-Helix 1653 cm^−1^	β-Sheets 1623 cm^−1^	β-Sheets 1637 cm^−1^	Random Coils 1645 cm^−1^	β-Turns 1663 cm^−1^	β-Turns 1694 cm^−1^
US effect						
10 min	1.32	1.08	0.79	1.19	1.38	1.00
20 min	1.17	1.29	1.20	1.74	3.11	1.21
30 min	0.88	1.18	1.01	1.83	1.55	1.40
40 min	0.96	1.18	1.19	1.36	1.77	1.22
IR effect						
3 kGy	0.20	0.74	0.70	1.68	1.59	0.74
5 kGy	0.96	0.70	0.48	1.35	1.35	0.42
7 kGy	0.18	0.67	0.55	1.66	1.08	0.50
US-IR effect						
US-IR	1.02	1.07	1.00	3.47	1.70	0.55

US: ultrasound treatment (10, 20, 30 and 40 min), IR: γ -irradiation treatment (3, 5 and 7 kGy), US-IR: IR assisted with US.

**Table 2 foods-10-02259-t002:** Percentage of molecular weight (MW) distribution during digestion (0 to 240 min) of non-fermented (NF) and fermented (F) beverages non-enriched (Control), untreated cricket protein-enriched beverages (Cr) and those treated with different processes (US-IR, US-E, US-EWC and US-IRE). Data shown are means ± SD (n = 3).

Percentage of MW Distribution, (%)									
Samples	MW (Da)	NF ^1^ (0 min)	NFP ^2^ (120 min)	NFPT ^3^ (240 min)	Δ_NF_^7^	F ^4^ (0 min)	FP ^5^ (120 min)	FPT ^6^ (240 min)	Δ_F_ ^7^
Control	>3000	6.7 ± 0.3	3.9 ± 0.1	2.2 ± 0.0	−67.1	6.4 ± 0.0	4.6 ± 0.2	2.3 ± 0.1	−62.5
	3000–260	60.0 ± 0.5	61.6 ± 0.5	63.6 ± 0.8	6.0	59.1 ± 1.1	60.0 ± 0.4	63.4 ± 0.6	7.6
	<260	33.3 ± 0.9	34.5 ± 0.4	34.2 ± 0.7	2.7	34.5 ± 1.0	35.4 ± 0.1	34.3 ± 0.6	−0.6
Cr	>3000	5.0 ± 0.61	3.8 ± 0.41	3.2 ± 0.35	−36.0	4.9 ± 0.7	3.8 ± 0.4	3.0 ± 0.4	−38.7
	3000–260	37.7 ± 2.9	38.8 ± 0.2	43.3 ± 0.1	17.9	41.4 ± 0.8	43.3 ± 0.6	45.5 ± 0.5	10
	<260	57.3 ± 3.2	57.4 ± 0.5	53.5 ± 0.3	−6.6	53.7 ± 0.0	52.9 ± 0.2	51.5 ± 0.0	−4.1
US-IR	>3000	4.5 ± 0.1	3.0 ± 0.1	3.3 ± 0.4	−26.6	4.5 ± 0.1	3.7 ± 0.2	3.5 ± 0.1	−22.2
	3000–260	37.1 ± 0.7	39.3 ± 0.3	49.1 ± 0.3	32.3	44.7 ± 1.61	43.1 ± 1.6	46.0 ± 0.4	2.9
	<260	58.4 ± 0.6	57.7 ± 0.2	47.6 ± 0.4	−18.4	50.8 ± 1.4	53.2 ± 1.5	50.5 ± 0.5	−0.6
US-E	>3000	1.8 ± 0.0	1.7 ± 0.1	0.9 ± 0.2	−50.0	2.2 ± 0.7	1.3 ± 0.0	0.9 ± 0.7	−59.0
	3000–260	38.7 ± 0.3	39.4 ± 0.0	41.7 ± 0.4	7.7	39.1 ± 0.4	40.5 ± 0.3	41.4 ± 1.2	5.9
	<260	59.5 ± 0.3	58.9 ± 0.1	57.4 ± 0.4	−3.5	58.7 ± 0.3	58.2 ± 0.2	57.7 ± 0.1	−1.7
US-EWC	>3000	1.3 ± 0.6	0.8 ± 0.0	0.5 ± 0.0	−61.5	1.8 ± 0.3	0.7 ± 0.0	0.5 ± 1.0	−72.2
	3000–260	29.4 ± 0.3	30.0 ± 0.0	30.5 ± 0.3	3.7	30.1 ± 0.3	31.1 ± 0.0	34.7 ± 2.5	15.3
	<260	69.3 ± 0.4	69.1 ± 0.5	69.0 ± 0.4	−0.5	68.2 ± 0.4	68.2 ± 0.2	64.8 ± 2.8	−4.9
US-IRE	>3000	1.3 ± 0.2	0.8 ± 0.5	0.7 ± 0.6	−85.7	1.3 ± 0.1	0.7 ± 0.4	0.3 ± 1.0	−76.9
	3000–260	18.9 ± 0.3	19.3 ± 0.4	18.5 ± 0.1	−2.1	20.4 ± 0.4	19.4 ± 0.0	19.0 ± 2.5	−6.8
	<260	79.8 ± 0.3	79.8 ± 1.0	80.7 ± 0.7	1.1	78.3 ± 0.4	79.8 ± 0.3	80.7 ± 2.8	3.1

^1^ NF: non-fermented, at the beginning of the digestion. ^2^ NFP: non-fermented, after the pepsin digestion. ^3^ NFPT: non-fermented, after the pepsin + trypsin digestion. ^4^ F: fermented, at the beginning of the digestion. ^5^ FP: fermented, after the pepsin digestion. ^6^ FPT: fermented, after the pepsin + trypsin digestion. ^7^ Δ: relative difference of MW distribution between the end and the beginning of the in vitro digestion of non-fermented beverages (Δ_NF_) and fermented beverages (Δ_F_). US-IR: treated with γ-irradiation assisted with ultrasound. US-E: treated cricket protein-enriched beverages subjected to enzymatic hydrolysis assisted with ultrasound. US-EWC: treated cricket protein-enriched beverages subjected to enzymatic hydrolysis assisted with ultrasound, whole product of hydrolysis. US-IRE: enzymatic hydrolysis assisted with γ-irradiation and ultrasound.

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
