# Peer review of "Effect of Physical and Enzymatic Pre-Treatment on the Nutritional and Functional Properties of Fermented Beverages Enriched with Cricket Proteins"

_foods, 2021, doi:10.3390/foods10102259_

Round 1

Reviewer 1 Report

The manuscript is interesting. Some improvements and minor corrections should be done, such as:

i) The initial protein content of the samples used in the study was characterised or the samples were protein free?  

ii) pg. 16, line 593: please replace " influenced by the fermenation" by " influenced by the fermentation".

Important: The combined treatment of cricket proteins proposed by authors is quite expensive. The authors should discuss about the added cost of this combined treatment for food or beverage industry.

Author Response

  1. Question/Comment:

The initial protein content of the samples used in the study was characterised or the samples were protein free?  

Answer: The initial protein content of the samples used in this study was characterized. Before adding cricket proteins, samples were contained 3% of rice proteins (non-enriched beverage).

Information have been mentioned on line 167, P 4.

  1. Question/Comment:
  2. 16, line 593: please replace " influenced by the used fermenation" by " influenced by the fermentation".

Answer: " influenced by the fermenation" have been replaced by " influenced by the fermentation". It is now on line 596.

  1. Question/Comment: The combined treatment of cricket proteins proposed by authors is quite expensive. The authors should discuss about the added cost of this combined treatment for food or beverage industry.

Answer: The combined treatment was carried out with the aim of increasing the process efficiency while ensuring the decrease in energy consumption and avoiding the restrictions due to a prolonged treatment duration. The use of enzyme is common for example for juice clarification. Gamma irradiation and ultrasound are considered as a less expensive processes compared to others physical processes (1,2). In our study, the use of combined treatments results in a very improved beverage in point of functional properties, particularly the solubility which is considered as the most important parameter for the beverage industry. Indeed, this parameter could influence not only the product acceptability by the consumer, but also the industrial feasibility (obtaining a smooth beverage, less viscous to avoid the risk of clogging of the installations included in the manufacturing process). In addition to the meeting the challenge of obtaining an enriched beverage with non-conventional proteins with nutritional properties (digestibility and peptide profile) equivalent or superior to other proteins of animal origin. Thereby, the added cost of pretreatment procedure is included in the cost of obtaining and improving of beverage quality.  Discussion have been added on line 606-…, P 17.

Reviewer 2 Report

This research paper is very interesting with great future (scientific and industrial) for the target groups mentioned in the introduction.

The abstract and introduction is adequate.

The content of the introduction is well documented. Materials and methods must be improved, particularly the statistical part.  In this point, authors should be more clear indicating what factors and levels each of the factors has and indicate why, having a multifactrial system, only is applied a ONEWAY-ANOVA analysis. 

Author Response

  1. Question/Comment:

Materials and methods must be improved, particularly the statistical part.  In this point, authors should be clearer indicating what factors and levels each of the factors has and indicate why, having a multifactorial system, only is applied a ONEWAY-ANOVA analysis. 

Answer: Only One Way Anova was used for the statistical analysis. Duncan’s multiple range tests for equal variances and Tamhane’s test for unequal variances were performed for statistical analysis.

Information have been added, line 281, 282.

Reviewer 3 Report

The presented manuscript for review is very interesting, it covers important issues of enriching food with non-animal protein. The work was planned correctly, however, a small supplementation is required in point 2.1 Materials, in which the source of the reagents used should be supplemented and it would be good if information appeared when organic Cricket flour was purchased. Moreover, the research methodology was correctly selected. The summary of the results does not raise any objections as well as their elaboration. The conclusions are sufficient. 

Author Response

Reviewer 3                                                                                                                        

  1. Question/Comment:

The work was planned correctly, however, a small supplementation is required in point 2.1 Materials, in which the source of the reagents used should be supplemented and it would be good if information appeared when organic Cricket flour was purchased.

Answer: Information have been added, in paragraph 2.1 Materials. All the reagents used in this study were indicated with their source.

Additional information: Minor modifications have been made: Line 442,